# Epigenetic Landscape and Therapeutic Implication of Gene Isoforms of Doublecortin-Like Kinase 1 for Cancer Stem Cells

**DOI:** 10.3390/ijms242216407

**Published:** 2023-11-16

**Authors:** Landon L. Moore, Courtney W. Houchen

**Affiliations:** 1Department of Medicine, The University of Oklahoma Health Sciences Center, Oklahoma City, OK 73104, USA; landon-moore@ouhsc.edu; 2Department of Veterans Affairs Medical Center, Oklahoma City, OK 73104, USA; 3The Peggy and Charles Stephenson Cancer Center, Oklahoma City, OK 73104, USA

**Keywords:** epigenetics, DCLK1, cancer stem cells, drug targeting, gene isoforms

## Abstract

While significant strides have been made in understanding cancer biology, the enhancement in patient survival is limited, underscoring the urgency for innovative strategies. Epigenetic modifications characterized by hereditary shifts in gene expression without changes to the DNA sequence play a critical role in producing alternative gene isoforms. When these processes go awry, they influence cancer onset, growth, spread, and cancer stemness. In this review, we delve into the epigenetic and isoform nuances of the protein kinase, doublecortin-like kinase 1 (DCLK1). Recognized as a hallmark of tumor stemness, DCLK1 plays a pivotal role in tumorigenesis, and DCLK1 isoforms, shaped by alternative promoter usage and splicing, can reveal potential therapeutic touchpoints. Our discussion centers on recent findings pertaining to the specific functions of DCLK1 isoforms and the prevailing understanding of its epigenetic regulation via its two distinct promoters. It is noteworthy that all DCLK1 isoforms retain their kinase domain, suggesting that their unique functionalities arise from non-kinase mechanisms. Consequently, our research has pivoted to drugs that specifically influence the epigenetic generation of these DCLK1 isoforms. We posit that a combined therapeutic approach, harnessing both the epigenetic regulators of specific DCLK1 isoforms and DCLK1-targeted drugs, may prove more effective than therapies that solely target DCLK1.

## 1. Introduction

Modern curative treatments for cancer have yet to substantially improve overall survival, especially for cases with widespread metastatic disease. Even with groundbreaking advances in genomics, proteomics, and technology, we urgently need innovative platforms to revolutionize cancer therapy. The complexity of the processes and regulatory mechanisms leading to cancer is still not fully understood, especially regarding the relationship between genetics and epigenetics in the context of tumor development in specific organs. Traditionally, cancer has been viewed as the result of uncontrolled cell growth due to the activation of oncogenes and the deactivation of tumor-suppressing pathways [1]. However, new findings suggest that cancer-causing processes might start even before these genetic changes occur. Environmental factors like radiation, toxins, infections, and inflammation can cause early changes at the DNA level, even before mutations in oncogenes become evident [2,3]. These changes include alterations in DNA methylation due to reactive oxygen species (ROS) [4], inflammatory cytokines [5], and DNA repair mechanisms [6]. All these factors can influence the activity of genes that suppress or promote tumor growth (Figure 1). Moreover, epigenetic changes, which are alterations that do not change the DNA sequence but affect gene expression, are pivotal in forming cancer stem cells (CSCs). These cells are implicated in tumor diversity, resistance to treatment, recurrence, immune system evasion, and metastasis [7,8]. With the help of advanced molecular techniques, we now see an even more complex landscape of genetic and epigenetic interactions driving cancer. A particular point of interest in this landscape is genes that have isoforms, or variants, that can either promote or suppress tumor growth depending on which version is active. An example of this is the gene doublecortin-like kinase-1 (DCLK1), a well-recognized marker for CSCs [9,10,11] that is linked with aggressive cancer types and resistance to treatment [12,13]. DCLK1 has multiple isoforms, and recent research suggests that they might have contrasting roles in cancer and other diseases [14]. Furthermore, these isoforms may be modulated via differential DNA methylation [15]. In this review, we delve deeper into the epigenetic regulation of such genes, focusing on DCLK1 and its isoforms. We explore how changes in its DNA methylation can influence which isoform is produced. We also advocate for the exploration of treatments that target these epigenetic changes in combination with isoform-specific therapies as a potential new avenue in cancer treatment.

## 2. Epigenetics and Cancer Progression

Epigenetics studies how gene activity changes without altering the DNA sequence. This field has shed light on the vital role these changes play in cancer’s development and progression. This discussion focuses on modifications resulting in alternative gene forms, primarily through DNA methylation and histone modifications.

### 2.1. DNA Methylation in Cancer

DNA methylation involves adding a methyl group to the carbon 5 position of the cytosine ring, usually in a CpG context, and this modification often regulates gene activity [7,16]. In cancer, research into the DNA methylome, the genome-wide mapping of the DNA methylation of cytosine residues, revealed a central anomaly: a reduction in methylation throughout the genome (hypomethylation) yet increased methylation (hypermethylation) at many gene promoter regions [17,18]. Global DNA hypomethylation was one of the first epigenetic alterations identified in cancer cells and results in the aberrant activation of proto-oncogenes, thus contributing to carcinogenesis [17,19]. In contrast, promoter hypermethylation often leads to the transcriptional silencing of tumor suppressor genes, contributing to cancer progression [18]. Both of these outcomes result from the observation that DNA methylation generally leads to the repression of gene expression via the direct inhibition of transcription factor binding or through the recruitment of proteins known as methyl-CpG-binding domain (MBD) proteins, which further recruit chromatin remodeling proteins that compact the DNA and limit accessibility for transcription [20]. 

Key proteins involved in this methylation process include DNA methyltransferases (DNMTs), which add methyl groups, typically leading to gene silencing [21]. Specifically, DNMT1 maintains existing methylation patterns, while DNMT3A and DNMT3B handle new methylation [22]. Overactivity in these enzymes can cause the abnormal methylation patterns seen in cancer. Additionally, MBD proteins recognize and bind to methylated DNA, often causing gene repression [23], while ten-eleven translocation (TET) proteins can initiate DNA demethylation, with their imbalance linked to several cancers [24].

### 2.2. Interplay between Histone Modifications and DNA Methylation 

Histone modifications play a crucial role in gene regulation by altering chromatin structure and function [25]. For example, adding acetyl groups (acetylation) to histones often activates genes [26]. Meanwhile, adding methyl groups (methylation) can activate or inhibit genes, depending on the specific location. For example, the trimethylation of histone H3 at lysine 4 (H3K4me3) is often associated with active promoters, while trimethylation at lysine 27 (H3K27me3) is linked with gene repression [27]. This interplay between DNA methylation and histone modifications is sometimes referred to as the “epigenetic code” (reviewed in Kong et al. [28]). While DNA methylation can influence the histone modification state, the reverse is also true: histone modifications can affect the DNA methylation pattern. One of the best studied interactions on the influence of histone methylation on DNA methylation is the methylation of histone H3 at lysine 9 (H3K9me), which leads to further DNA methylations and gene silencing [29]. Additionally, another modification, H3K27me3, can lead to gene silencing and is typically seen in DNA regions that are also hypermethylated [30].

Lysine demethylases (KDMs) are enzymes that remove methyl groups from histones, influencing DNA methylation in cancer. For instance, KDM1A removes such groups from H3K4, generally associated with gene activation [31]. This enzyme works with DNMTs, potentially directing them to specific DNA regions, leading to methylation and gene silencing [32]. Other KDMs can erase marks that typically inhibit gene activity, promoting gene expression, and when these enzymes are disrupted or mutated, it can result in cancer [33].

### 2.3. Regulation of Alternative Promoters through Epigenetics

Alternative promoters play a crucial role in gene expression, often producing multiple transcripts from a single gene. The choice of promoter can be influenced by various factors like cellular environment, development stages, and external conditions [34,35]. This variability enables genes to produce protein isoforms that can have differing, or even contrasting, functions [36]. Many regulatory mechanisms, from epigenetic modifications like DNA methylation to cellular signaling, guide this process [37,38,39]. The misregulation of these processes can result in diseases, including cancer.

In human tissues, DNA methylation showcases the dynamic nature of promoter regulation [40]. In the context of cancer, changes in DNA methylation at promoter sites can lead to a diversity of gene isoforms and even serve as potential disease markers [41,42,43,44], with differences in the chromatin structure at alternative promoters likely responsible for the dysregulation observed in cancers [45]. Often, one promoter is rich in CpG sites, while another is not. The CpG-deficient sites often undergo demethylation, leading to gene activation [44]. Ensuring methylation maintenance is essential for proper promoter regulation. For example, the absence of DNMT3B, a protein linked to methylation issues in cancer, can trigger the unwanted activation of alternative promoters [46,47,48].

### 2.4. Epigenetics and Alternative Splicing

Alternative splicing, influenced by promoter usage, results in various gene isoforms [49,50]. The pattern of DNA methylation can impact this splicing, directing the creation of specific gene versions. Any change in methylation near splicing sites can alter the binding patterns of the splicing machinery, leading to varied outcomes [51]. Histone modifications also play a role; for instance, methylation-induced changes in chromatin structure can affect the transcription process and splicing decisions. The acetylation of histones opens the chromatin, which can influence the recognition of splice sites [52,53]. Overall, the speed of RNA polymerase II and the assembly of spliceosomes are affected by these epigenetic changes, altering splicing results [54]. In summary, epigenetics—especially DNA methylation and histone changes—profoundly affect cancer development. These mechanisms regulate gene activity without changing the DNA structure, either activating cancer-causing genes or silencing those that suppress tumors. Through alternative promoters and splicing, they drive the creation of diverse gene forms that could influence cancer’s progression.

## 3. Gene Isoforms in Cancer: A Double-Edged Sword

Gene regulation becomes more intricate due to alternative promoters, leading to diverse protein isoforms that significantly sway cell behavior. This phenomenon, known as promoter choice, results in the creation of unique mRNA and protein isoforms. These isoforms can sometimes perform contrasting functions [34]. Cancer development relies on the balance between oncogenes, which promote cell growth, and tumor suppressors, which restrain uncontrolled cell division. Notable oncogenes include MYC, KRAS, and EGFR, while TP53, BRCA1/2, and PTEN are well-known tumor suppressors [1,55]. This balance is foundational to understanding cancer’s molecular nature and shapes the design of therapies targeting these proteins [56].

Tumorigenesis is multi-faceted, characterized by both genetic and epigenetic changes that lead to unregulated cell growth. One pivotal component in this complex landscape is gene isoforms, which are various forms of the same gene. Surprisingly, some genes can yield isoforms that act as both oncogenes and tumor suppressors. For instance, members of the p53 family, *TP63* and *TP73*, produce TAp63/TAp73, which resemble tumor suppressors, and ΔNp63/73, which might have oncogenic properties [57,58]. Similarly, the *RASSF1* gene presents RASSF1A as a tumor suppressor and RASSF1C with potential oncogenic roles [59]. The regulation of these isoforms can be influenced by epigenetic modifications, introducing another layer of intricacy in their roles in cancer [8].

One key protein, DCLK1, known to regulate CSCs, is gaining attention in cancer research (reviewed by Chhetri et al. [60]). It is linked to the self-renewal and tumorigenic abilities of CSCs in various cancers [61,62,63]. Epigenetic mechanisms, like DNA methylation and histone modifications, can regulate the expression of *DCLK1* and the stem-like properties of CSCs, affecting tumorigenesis and treatment responses [64]. A growing body of evidence suggests that DCLK1 is a marker for CSCs and a promising target for therapies aiming to halt tumor recurrence and metastasis [65,66,67]. The presence of alternative promoters complicates gene regulation, resulting in a plethora of protein isoforms pivotal in cancer development. This intricate dance between oncogenes and tumor suppressor genes is central to the molecular essence of cancer and is invaluable for therapeutic development.

## 4. Epigenetic Regulation of Alternative Isoforms in Cancer

Gene isoforms can function as either oncogenes or tumor suppressors, based on their context and differential expression. Epigenetic regulation, notably DNA methylation and histone modifications, often governs the expression of these isoforms. Examples include *RASSF1*, *TP73*, and *DCLK1* genes [68].

### 4.1. RASSF1

RASSF1 plays a key role in cell cycle regulation and has implications in various cancers. The gene yields two main isoforms, RASSF1A and RASSF1C, through alternative promoter usage [59,69]. RASSF1A, a known tumor suppressor, is often silenced in many malignancies [70]. Additionally, the *RASSF1* promoter methylation status serves as a diagnostic and prognostic marker in breast, lung, and ovarian cancer [71,72,73]. On the other hand, RASSF1C is believed to have oncogenic potential [59]. Importantly, the two isoforms differ in a protein domain that influences their role in apoptosis and cellular proliferation. The RASSF1 5′ promoter responsible for RASSF1A contains a CpG island, which, when hypermethylated, leads to transcriptional silencing of the gene [69], while, in contrast, the downstream promoter that generates RASSF1C is less frequently methylated.

The fundamental difference between RASSF1A and RASSF1C is that RASSF1A contains a protein kinase C domain lacking in RASSF1C [59]. One consequence is that RASSF1C does not associate with death receptor complexes the same way as RASSF1A, leading to the inhibition of apoptosis and promotion of the upregulation of proliferation and invasive phenotypes [74]. Thus, these divergent roles are due to the absence of a single domain that is important in regulating physical interactions [75,76].

### 4.2. TP63/TP73

This theme of one isoform lacking a functional domain that determines its role in cancer is also seen in the *TP63/TP73* gene that also has multiple gene isoforms that play a crucial role in multiple cellular functions including differentiation, proliferation, and apoptosis. Both *TP63* and *TP73* resemble the well-studied *p53* tumor suppressor gene but exhibit structurally and functionally distinct isoforms arising from alternative promoters and extensive alternative splicing [77]. In both TP63 and TP73 gene isoforms, the upstream promoter (P1) drives the transcription of full-length or transactivating (TA) isoforms, which have a similar structure to p53 and encompass transcriptional activation, DNA binding, and oligomerization domains [57]. The TA isoforms often function as tumor suppressors, promoting cell cycle arrest and apoptosis. On the other hand, the alternative internal promoter (P2) drives the expression of N-terminally truncated (∆N) isoforms that lack the transcriptional activation domain and act in a dominant-negative manner, inhibiting the function of TA isoforms and promoting tumorigenesis [57]. Furthermore, CpG methylation within the P1 promoter region has been reported, resulting in transcriptional repression of the TA isoforms [78,79].

### 4.3. DCLK1

*DCLK1* presents several isoforms due to alternative promoters and mRNA processing. Although their specific roles in cancer are yet to be fully understood, DCLK1.4 seems to be particularly crucial. Unlike TP63/TP73 where loss of the transcriptional activation domain allows for a dominant negative effect, DCLK1 is not known to form multimers; however, DCLK1 may be more like RASSF1 in that the different isoforms interact with different complexes to modulate their activity. DCLK1 is associated with CSCs [9,10,11] and promotes characteristics linked with aggressive cancer [12,13]. DCLK1 kinase inhibitors have shown promise in reducing tumorigenesis [80,81,82]. However, antibodies directed against a non-kinase extracellular domain also showed significant effects on tumor progression [83,84]. These results suggest that DCLK1 may also have alternative domains present in distinct isoforms that define its role in cancer progression.

DCLK1 was first described as a brain-specific protein with similarity to the previously identified doublecortin (DCX) gene [85,86,87,88]. Both DCLK1 and DCX have a pair of domains required for tubulin binding and microtubule assembly [89,90]. However, DCLK1 also has a domain resembling the Ca^2+^/calmodulin-dependent protein kinase indicating additional functionality [87,91]. The *DCLK1* locus generates several isoforms using alternative promoters and alternative mRNA processing (Figure 2) [14]. Interestingly, DCLK1 can be proteolytically processed into a shorter form lacking the microtubule binding domains, indicating a possible functional difference between DCX-containing and non-DCX fragments of the protein [92,93]. Moreover, DCLK1 isoforms are differentially expressed and localized in developing mouse brains [94]. This study also showed that DCLK1 isoforms were generated via transcription, indicating regulation by distinct promoters. Furthermore, a study using isoform-specific antibodies found that the different isoforms tended to accumulate in different locations, suggesting that localization is important for distinctive functions [95]. Given that different *DCLK1* promoters generate unique isoforms and that these isoforms have altered functional domains, it is likely that the DCLK1 isoforms have unique functions. 

DCLK1 is transcribed from two distinct promoters [14,15]. The 5′ most promoter, termed the alpha-promoter (DCLK1_alpha_), produces two full-length transcripts encoding proteins that contain both the microtubule-binding doublecortin domains and the Ca^2+^/calmodulin-dependent protein kinase domain, often referred to as DCLK1-long (DCLK1-L). However, alternative processing, whereby a specific exon is either retained or skipped, distinguishes DCLK1-L into two isoforms that differ in their C-termini [14]; we refer to DCLK1-L with the exon retained as DCLK1.1 and the variant with the exon skipped as DCLK1.2. Because the alternative processing leads to a change in the respective open reading frame, DCLK1.2 is longer than DCLK1.1 and is different in its final C-terminal sequence, providing a way to distinguish between these two isoforms (Figure 2). Furthermore, the expression of DCLK1.1 and DCLK1.2, being driven by the same promoter, at least transcriptionally, is coordinated. Studies in colorectal, pancreatic, gastric, and lung cancers have indicated that this promoter may be epigenetically silenced through hypermethylation [13,15,96,97]. In cases of DCLK1_alpha_ promoter silencing, DCLK1 is still observed due to a second promoter, termed the beta promoter (DCLK1_beta_), present downstream of the DCLK1_alpha_ promoter and located in an intron. This DCLK1_beta_ promoter produces transcripts that are translated into proteins lacking the microtubule-binding domain and are termed DCLK1-short (DCLK1-S) [14,98]. The same potential for alternative splicing as in DCLK1-L occurs in DCLK1-S [98], and we refer to these two isoforms as DCLK1.3 (retained exon) and DCLK1.4 (skipped exon). Furthermore, DCLK1.4 was independently isolated in rats as the candidate plasticity gene (CPG16) supporting the conclusion that the DCLK1_beta_ promoter and its products are biologically relevant [91]. The DCLK1_beta_ promoter lacks prominent CpG islands with which to be silenced and appears that it may be regulated by NF-kBp65 [15] and FOXD3 [99]. Furthermore, the DCLK1_beta_ promoter is also regulated by lymphoid enhancer-binding factor (LEF1) [100]. However, it remains unclear whether promoter usage regulates the alternative splicing or that the alternative splicing itself is subject to regulation to generate these DCLK1 isoforms. 

Numerous studies have implicated the high expression of DCLK1 with poor prognosis and that the inhibition of DCLK1 via a variety of methods suggests that targeting DCLK1 is a viable strategy for cancer therapy [60]. Recent work has focused on the role of different isoforms of DCLK1 in cancer progression. Unfortunately, most of these studies did not distinguish between the various DCLK1 isoforms (see Kalantari et al. [101]). What is identifiable is that DCLK1 modulates several pathways important in cancer progression, such as the epithelial-to-mesenchymal transition (EMT), cancer stemness, inflammation, and metastasis. Furthermore, the overexpression of DCLK1.2 was seen to increase cancer aggressiveness in PDAC and that targeting DCLK1.2’s unique C-terminal domain could inhibit tumorigenesis [102]. It has also been linked to tumor immunosuppression via M2-macrophage polarization [103]. An important caveat to these types of overexpression studies with kinases is that these studies represent unfettered kinase activity that may obscure the subtle aspects of kinase regulation [104]. 

Knockdown studies and DCLK1 kinase inhibitor studies have shown improvements in reducing tumorigenesis and modulating the tumor microenvironment [105]. However, most of these studies are broad-spectrum, affecting all isoforms of DCLK1 as they all retain the kinase domain, raising the question of whether dysregulated or overactive DCLK1 kinase is the root cause. Interestingly, treatments targeting the C-terminal domain of DCLK1.2 and DCLK1.4 show significant effects on cancer progression, suggesting that the C-terminal domain is important [84]. It is not uncommon for kinases to be regulated by a C-terminal domain and, indeed, a C-terminal autoinhibitory domain (AID) common to all isoforms was identified that regulates DCLK1 autophosphorylation [104,106]. However, an antibody CBT-15 against the unique C-terminal domain of DCLK1.2/DCLK1.4 outside of the AID domain in the intrinsically disordered domain could inhibit tumor progression, indicating the existence of other regulatory domains [83,84]. 

Recent work has focused on the oncogenic role of the DCLK1-S isoform [99,107]. DCLK1.4 is detected in CSCs, suggesting a role in their formation and/or maintenance [95,100]. Furthermore, the inhibition of DCLK1.4 transcription via the DCLK1_beta_ promoter through blocking LEF-1 diminishes cancer stemness [100]. DCLK1.4 also induces the EMT, a key property related to the increased aggressiveness of cancer and immune suppression [108,109]. DCLK1 regulates inflammation in both human and murine colitis with DCLK1.2 and DCLK1.4 showing differential regulation [68]. Additionally, these changes were correlated with a decrease in FoxD3, an inhibitor of the DCLK1_beta_ promoter, suggesting that DCLK1.4 upregulation acts to promote inflammation, a known driver of tumorigenesis, and that the balance between DCLK1.2 and DCLK1.4 is important. In colorectal cancer, DCLK1.4 was shown to promote cancer stemness and aggressiveness, consistent with its overexpression relative to DCLK1.2 [9,15]. This was dependent on the DCLK1 phosphorylation of XRCC5 and the co-option of the inflammatory tumor microenvironment (TME) [9]. Given the results that DCLK1.4 isoform overexpression is highly oncogenic, while DCLK1.2, through its silencing, fits the definition of a tumor suppressor, the restoration of DCLK1.2/DCLK1.4 balance most importantly appears to be vital for blocking cancer progression.

## 5. The Interplay of Gene Isoforms, Epigenetics, and Cancer Stem Cells

Epigenetic regulation holds a key role in the dynamics of CSCs, a subset of cells within tumors responsible for tumor initiation, progression, therapy resistance, and relapse [7,8]. This regulation, primarily through DNA methylation and histone modifications, influences the balance between CSC self-renewal, differentiation, and oncogenic potential, making it vital in the cancer context [110]. In CSCs, a range of gene isoforms under the control of their specific epigenetic environments are produced [111]. For example, CD44, a protein linked to CSCs, has various isoforms influenced by DNA methylation. When hypermethylated, there is a shift from expressing the standard CD44s isoform to the CD44v variant, which is associated with a more aggressive cancer type [112,113].

*DCLK1* is another key gene impacted by epigenetic modifications. The *DCLK1* locus shows methylation of the cytosines in CpG islands overlapping the alpha promoter in multiple cancers, including colon cancers [13,15,96,114]. This methylation results in silencing of the alpha-promoter and inhibits the expression of the DCLK1-L isoforms. However, increased DCLK1 expression is observed in colon cancers, suggesting that transcription from the beta-promoter and upregulation of the DCLK1-S isoforms are vital for cancer progression [15]. This silencing of one promoter may be necessary for the increased activity of the downstream promoter, thus emphasizing the role of epigenetic controls in isoform generation.

Lysine demethylase 3A (KDM3A), a histone demethylase, is an epigenetic modulator that plays a significant role in multiple cellular processes and is implicated in the regulation of gene isoforms. By modifying histone H3, KDM3A can impact gene transcription and potentially alternative splicing. KDM3A demethylates histone H3 on lysine 9 (H3K9), enhancing gene transcription [115]. While its specific role in alternative splicing and isoform regulation is still unclear, it is known that alterations in histone modifications, such as those catalyzed by KDM3A, can influence the splicing machinery and impact the balance of gene isoforms [52]. In cancer, KDM3A is often overexpressed and associated with poor prognosis [116]. Interestingly, KDM3A also appears to regulate specific DCLK1 isoforms, suggesting a deeper connection between histone modifications and isoform control [117]. With KDM3A being a promising therapeutic target, its inhibitors are currently under development (reviewed in Das et al. [118]).

In summary, the synergy between gene isoforms and epigenetics is fundamental in CSC behavior and therapy resistance. Insights into their dynamics, such as the differential expression of CD44 and DCLK1 isoforms or the regulatory role of KDM3A, can provide potential therapeutic strategies for more effective cancer treatments. 

## 6. The Future of Cancer Treatment: Isoform-Specific Therapies and Epigenetic Modulation

Cancer therapeutics has undergone a paradigm shift over the last few decades, moving from generalized cytotoxic agents toward more targeted therapies including those directed at the epigenome [119]. A promising avenue in this realm is the targeting of specific gene isoforms and using compounds that modulate epigenetic changes to favor certain isoforms (Figure 3). Many proteins possess various isoforms, each with unique cellular roles. For example, the phosphatidylinositol-3-kinase (PI3K) protein, vital in cell growth and division, has several isoforms [120]. Some of these become aberrant in certain cancers, making them potential therapeutic targets. The FDA-approved PI3K inhibitor, Idelalisib, which targets a specific PI3K isoform, has shown efficacy against blood cancers [121]. Likewise, trastuzumab targets an overexpressed isoform in particular breast cancers [122]. A challenge in this therapeutic strategy is the structural similarities between protein isoforms, which might result in unintended side-effects. Yet, technological advancements allow for better-targeted drugs. For instance, certain strategies are in development to specifically target the oncogenic RASSF1C isoform [123]. In the case of DCLK1, all isoforms retain the active kinase domain, so targeting that domain will affect all isoforms. Thus, several drugs have been either repurposed or directly developed to target the DCLK1 kinase domain (Table 1). However, a DCLK1-isoform-specific treatment would be more useful in that it could target only the isoforms that are involved in cancer stemness and progression. Currently, no inhibitors of the microtubule binding domain exist. Therefore, we developed a strategy to target the unique C-terminal domain of the DCLK1.4 and DCLK1.2 isoforms, which are implicated in cancer progression, to provide for a more direct isoform-specific treatment method [84]. We generated antibodies that target the unique C-terminal domain of DCLK1.2/DCLK1.4. These antibodies along with the chimeric antigen receptor T-cells (CAR-T) created from them bind to a distinct epitope present only in these isoforms and have demonstrated the ability to kill cancer cells and prevent the growth of cancerous tumors in animal models, as shown in our study [83] (see Table 1). Additionally, we confirmed that this C-terminal domain is located on the cell’s exterior because we observed antibody binding to living cell surfaces, which aligns with other research. However, the mechanism by which DCLK1 is positioned on the cell surface is uncertain due to the absence of a known transmembrane domain.

The regulation of gene isoforms through epigenetic modulation offers another therapeutic angle. Specifically, targeting the methylation responsible for repressing the upstream promoter may aid in restoring the balance between the gene isoforms. Such therapies may take the form of inhibition of the DNA methylases involved in maintaining the methylation status directly, i.e., DNMT1, or the histone modifications that control the local chromatin structure to promote demethylation. Major drawbacks on using such therapies are the lack of locus specificity, resulting in global and non-chromatin effects; i.e., the targets for these compounds are often essential proteins, like DNMT1 or KDM1 (Table 2). As such, these drugs are not used alone but are often used in combination with other drugs (reviewed in Majchrzak-Celińska et al. [128] and Sahafnejad et al. [129]). As such, natural compounds like curcumin, resveratrol, and epigallocatechin-3-gallate are highly prospective drugs that can influence epigenetic shifts, favoring tumor suppressor isoforms (Table 2).

Curcumin from turmeric influences DNA methylation and histone modifications to induce the hypomethylation of DNA, leading to the reactivation of tumor suppressor genes [138] and altering cell proliferation and apoptosis [139]. These effects of curcumin on epigenetic processes show promise for a role in cancer therapeutics; however, curcumin has shown mixed effects in cancer studies. While curcumin reduced the expression of DCLK1 overall, a subset of cells was resistant, and resistance was abrogated through simultaneous knockdown of DCLK1 [140]. This result suggests that while curcumin was effective in reducing DCLK1 levels, DCLK1 total expression is not the leader but the balance of isoforms may act as the driving force. 

In addition to curcumin, there are several other compounds, many of which are naturally occurring, that are known to regulate epigenetic modifications and are being studied for their potential use in cancer therapy. For example, other natural compounds, like resveratrol, EGCG, sulforaphane, genistein, and quercetin, have all demonstrated potential in modifying epigenetic processes, thereby influencing cancer cell behavior [138,139]. Resveratrol, found in grapes, red wine, and berries, has been shown to affect various epigenetic processes [141,142,143]. EGCG [134], a major active component of green tea; sulforaphane [144], found in cruciferous vegetables; genistein [145], a soy isoflavone; and quercetin [146], a flavonoid found in many fruits and vegetables, have all shown potential anticancer activity by affecting epigenetic processes via regulation of DNA methylation and histone modification in cancer cells.

A significant challenge in epigenetic cancer therapy is chemoresistance, often attributed to the properties of cancer stemness [147,148]. To overcome this, a combined approach targeting both CSCs and epigenetic alterations may be effective. DCLK1 is a recognized marker essential for cancer stemness [9,11,67,80,100,149,150,151]. Therefore, therapies that target DCLK1, either broadly using DCLK1 kinase inhibitors or more precisely with isoform-specific treatments, alongside epigenetic drug therapy, could offer a promising strategy to reestablish the balance between oncogene and tumor suppressor isoforms in cancer.

The intersection of isoform-specific targeting and epigenetic modulation holds immense potential for future cancer therapies. Therapies aimed at restoring the expression of tumor suppressor isoforms or inhibiting oncogenic isoforms could provide novel strategies for treating cancer [152]. The challenge, however, lies in the targeted delivery and specificity of such interventions. By profiling the epigenetic landscape of a patient’s tumor, one may be able to predict disease course, resistance patterns, and optimal therapeutic strategies [129]. As we deepen our understanding of the relationship between gene isoforms, epigenetics, and CSCs, the potential for tailored treatments grows. Profiling the epigenetic attributes of a patient’s tumor might enable predictions about the disease’s progression, therapeutic resistance, and most effective treatments.

The future of cancer treatment is veering toward tailored approaches, leveraging the power of isoform-specific drugs, as demonstrated by drugs like Idelalisib, and epigenetic modulators, such as curcumin and resveratrol, hold promise. It is the nuanced targeting of specific protein isoforms that stands out as a transformative approach in cancer treatment. As research continues in this direction, the hope for more personalized and effective treatments becomes increasingly tangible.

## 7. Conclusions

Epigenetics plays a pivotal role in governing gene isoforms and has steered scientific interest toward combined therapies that employ isoform-specific drugs alongside epigenetic modulators. Emerging evidence indicates that CSCs exhibit a heightened susceptibility to epigenetic alterations. Therefore, a dual-pronged approach targeting both the unique isoforms and the overarching epigenetic landscape could revolutionize cancer treatments. DCLK1, an alternative promoter gene implicated in tumorigenesis, exemplifies the challenges and opportunities in this domain. Despite its multiple isoforms, the kinase domain remains consistent, hinting that the diverse functions might hinge on specific localization or kinase activity regulation. Consequently, modulating the balance of DCLK1 isoforms might be more efficacious than targeting a particular isoform. However, targeting a specific isoform might be necessary. Encouragingly, recent research that zeroes in on the distinct C-terminus of some DCLK1 isoforms has shown potential in countering cancer traits. This suggests that strategies pivoting away from the kinase domain could bear fruit. Yet, a consistent preference for a single DCLK1 isoform across all cancers remains elusive. This inconsistency underscores the need for a deeper dive into the distinct roles of each DCLK1 isoform.

## Figures and Tables

**Figure 1 ijms-24-16407-f001:**
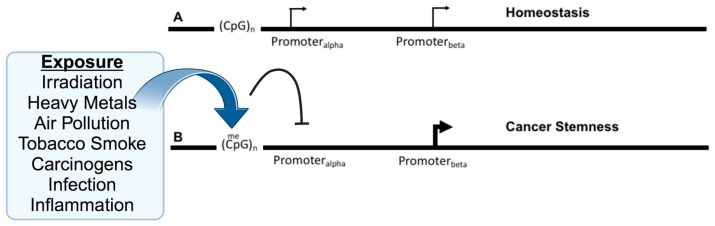
Exposure to environmental insults, such as irradiation or inflammation, alter DNA methylation and gene isoforms to promote cancer. (**A**) Several genes have been identified as having alternative promoters, Promoter_alpha_ and Promoter_beta_, with the upstream promoter located near CpG-islands that are the target of DNA methylation. Expression may be primarily from the upstream promoter, but this does not preclude expression from the downstream promoter, which may reflect the homeostatic balance of the individual isoforms and their roles. (**B**) Upon various environmental exposures, the DNA methylation of the upstream promoter is hypermethylated at nearby CpG sites resulting in the inhibition (arc with crossbar) of the promoter. This may directly or indirectly result in an increase in expression from the downstream promoter, which may affect tumorigenesis via the imbalance between the isoforms and their relative functions. Created with BioRender.com (accessed on 10 October 2023).

**Figure 2 ijms-24-16407-f002:**
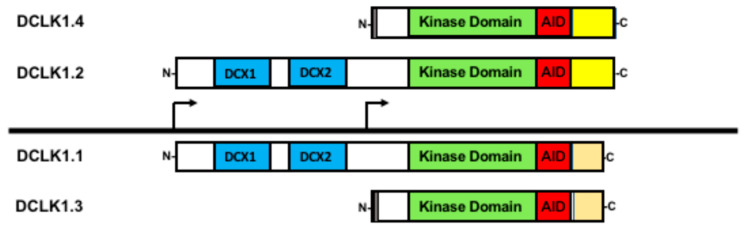
Doublecortin-like kinase-1 (DCLK1) isoforms generated via alternative promoter and alternative RNA splicing. All the four main isoforms contain the kinase domain (green) and its autoregulatory domain (AID) shown in red. Isoforms DCLK1.1 and DCLK1.2 (often referred as DCLK1-L) both contain the microtubule binding domains (DCX, shown in blue) yet differ in their C-terminal domains due to alternative RNA processing (yellow versus orange). From the downstream promoter, two isoforms (sometimes referenced as DCLK1-S) are generated also with distinct C-termini from alternative RNA processing, providing a further means of functional regulation. A short sequence at the N-terminus unique to the two downstream promoter products is shown in grey. Created with BioRender.com (accessed on 10 October 2023).

**Figure 3 ijms-24-16407-f003:**
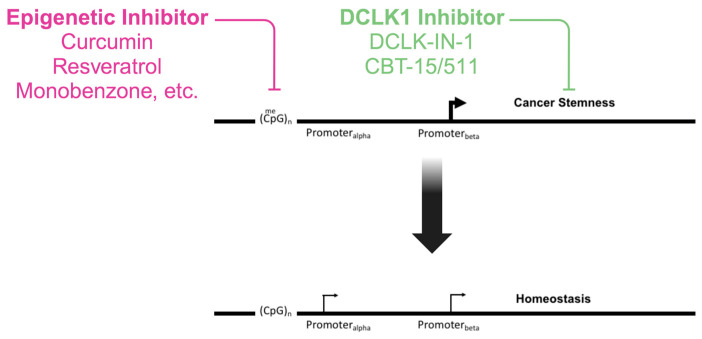
Proposed model for combined epigenetic drug therapy along with DCLK1 targeted inhibition of cancer stemness to restore homeostasis. Epigenetic inhibitor drugs, preferably with minimal side-effects, are used to oppose the epigenetic marks, inhibiting the expression of tumor suppressors and oncogene-opposing isoforms. Simultaneously, the inhibition of DCLK1, either isoform-specific (CBT-15/511) or in general, by targeting the kinase activity (DCLK1-IN-1) is used to block cancer stem cells that promote chemoresistance. Created with BioRender.com (accessed on 10 October 2023).

**Table 1 ijms-24-16407-t001:** Known DCLK1 inhibitors and their targeted domain.

Drug	Target	Type	Reference
XMD-17-51	Kinase Domain	Small molecule	Yang et al., 2021 [124]
DCLK1-IN-1	Kinase Domain	Small molecule	Ferguson et al., 2020 [82]
Ruxolitinib	Kinase Domain	Small molecule	Jang et al., 2021 [125]
XMD8-92	Kinase Domain	Small molecule	Sureban et al., 2014 [126]
LRRK2-IN-1	Kinase Domain	Small molecule	Weygant et al., 2014 [127]
CBT-15/CBT-511	Unique C-terminus of DCLK1 isoforms 2/4	Monoclonal antibody/CAR-T	Sureban et al., 2019 [83]

**Table 2 ijms-24-16407-t002:** Potential epigenetic drugs for restoring isoform balance.

Drug	Target	FDA Approved	Reference
Azacitidine	DNMT1	Yes	Kaminskas et al. [130]
Decitabine	DNMT1	Yes	Erdmann et al. [131]
Zebularine	DNMT1	No	Balch et al. [132]
Monobenzone	KDM1	No	Ma et al. [133]
Epigallocatechin gallate	Broad	No	Li et al. [134]
Curcumin	Broad	No	Sultana et al. [135], Mohamadian et al. [136]
Resveratrol	Broad	No	Chatterjee et al. [137]

## Data Availability

Not applicable.

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
