# Peer review of "Epigenetic Landscape and Therapeutic Implication of Gene Isoforms of Doublecortin-Like Kinase 1 for Cancer Stem Cells"

_ijms, 2023, doi:10.3390/ijms242216407_

Round 1

Reviewer 1 Report

Comments and Suggestions for Authors

The review is titled, "Dual Role of Gene Isoforms of DCLK1 in Tumorigenesis: Epigenetic Regulation and Implication for Cancer Stem Cell". However, the dual role of different isoforms of DCLK1 is not clearly laided out. A large portion is about DNA methylation, histone modifications, alternative promoter usage, and alternative splicings, which are the topics that have been extensively reviewed before. Even Tp63/TP73 and RASSF1 isoforms are heavily talked about, which is not the topic of this review. I think there just is not enough known about DCLK1 and isoforms to justify a review now.

Comments on the Quality of English Language

Not a big concern, except a few complex sentences should be simplified, such as, "However, it’s not just about inhibition of specific isoforms is
not the only approach, promoting certain isoforms is an alternative, albeit more complex, strategy where epigenetic modulators come into play"

Author Response

REVIEWER 1
The review is titled, "Dual Role of Gene Isoforms of DCLK1 in Tumorigenesis: Epigenetic Regulation and Implication for Cancer Stem Cell". However, the dual role of different isoforms of DCLK1 is not clearly laided out. A large portion is about DNA methylation, histone modifications, alternative promoter usage, and alternative splicings, which are topics that have been extensively reviewed before. Even Tp63/TP73 and RASSF1 isoforms are heavily talked about, which is not the topic of this review. I think there just is not enough known about DCLK1 and isoforms to justify a review now. 
We have revised the current manuscript to convey better the scope and focus of this review article for the special issue on epigenetics and drug therapies.

Reviewer 2 Report

Comments and Suggestions for Authors

Dual Role of Gene Isoforms of DCLK1 in Tumorigenesis: Epigenetic Regulation and Implication for Cancer Stem Cells

Overview

DCLK1, a protein kinase target for cancer therapy is gaining interest. It is a CSC marker which is highly expressed in various cancers. However, there are currently no therapeutic candidates in clinical studies that target DCLK1 kinase. This review provides the basic information and paves the way to genetic and epigenetic regulation associated with DCLK1.

Major Comments

1.     Authors stated “we investigated drugs that epigenetically regulate isoform specific generation with the hypothesis that a combination therapy that targets the epigenetic regulators of isoform specific DCLK1 with anti-DCLK1 drug will provide for more effective therapies than broadly anti-DCLK1 therapies”. However, no such information is provided in the text. Please include the drugs (include clinical trials also) which can target DCLK1.

2.     Please add few more figures and tables for better readability.

3.     4. References, only 35 references used are from recent literature (on or after 2020). Please use more recent articles.

Minor Comments

1.     Line 56: DCLK1 is already abbreviated in abstract, Line 16, please use abbreviation only.

2.     Abbreviations should be indicated at the first mention and used thereon. For example, Line 395, CSCs is already abbreviated in Line 47.

3.     Please increase the size of figures.

4.     Abbreviations should only be used, if they are being mentioned more than thrice. For example, EGCG, Line 436, only used twice. No need to use the abbreviation.

Remark

The review is well written and organized. Only few minor adjustments are needed.

Comments on the Quality of English Language

Few formatting mistakes should be rectified, otherwise language presentation is ok.

Author Response

REVIEWER 2
Major Comments 
Authors stated, “we investigated drugs that epigenetically regulate isoform specific generation with the hypothesis that a combination therapy that targets the epigenetic regulators of isoform specific DCLK1 with anti-DCLK1 drug will provide for more effective therapies than broadly anti-DCLK1 therapies”. However, no such information is provided in the text. Please include the drugs (include clinical trials also) which can target DCLK1. 
While there are not any current clinical trials using DCLK1 inhibitors, we present the case for examining such inhibitors in the context of combination therapy with epigenetic drugs.
Please add few more figures and tables for better readability. 
We have added an additional Figure 3 that illustrates the proposed combination drug therapy strategy. Additionally, we have improved prior figures for readability and clarity. Also, two tables are including with information relevant to drugs important for our discussion. We have provided additional references to current reviews focused on specific drugs, yet beyond our scope, so readers may delve deeper into the related drugs.
References, only 35 references used are from recent literature (on or after 2020). Please use more recent articles. 
We have included additional references more current when possible without taking credit away from the original works. 
Minor Comments 
Line 56: DCLK1 is already abbreviated in abstract, Line 16, please use abbreviation only. 
It is our understanding that the standard practice is to define abbreviations at first mention in both the abstract and in the general text. If this is the policy that abstract and general text are considered a continuation, then we will adjust as needed. Thanks for pointing this out.
Abbreviations should be indicated at the first mention and used thereon. For example, Line 395, CSCs is already abbreviated in Line 47. 
We apologize and have removed.
Please increase the size of figures.
We apologize for the error in uploading and have increased the size where needed. 
Abbreviations should only be used if they are being mentioned more than thrice. For example, EGCG, Line 436, only used twice. No need to use the abbreviation. 
We have removed it.